# New Approaches to Processing Ground-Based SAR (GBSAR) Data for Deformation Monitoring

**Sichun Long [1,2,3,\*], Aixia Tong [2], Ying Yuan [2], Zhenhong Li [3,\*]**  **, Wenhao Wu [1,2] and Chuanguang Zhu [1]**

[1] Hunan Key Laboratory of Coal Resources Clean-Utilization and Mine Environment Protection, Hunan University of Science and Technology, Xiangtan 411201, China; whwu@hnust.edu.cn (W.W.); cgzhu@hnust.edu.cn (C.Z.)

[2] School of Resource Environment and Safety Engineering, Hunan University of Science and Technology, Xiangtan 411201, China; anxia@hnust.edu.cn (A.T.); yyuan@huust.edu.cn (Y.Y.)

[3] School of Engineering, Newcastle University, Newcastle upon Tyne NE1 7RU, UK

[\*] Correspondence: sclong@hnust.edu.cn (S.L.); Zhenhong.Li@newcastle.ac.uk (Z.L.); Tel.: +86-13016178519 (S.L.); +44-779-089-0820 (Z.L.)

**Abstract:** In this paper, aiming at the limitation of persistence scatterers (PS) points selection, a new method for selecting PS points has been introduced based on the average coherence coefficient, amplitude dispersion index, estimated signal-to-noise ratio and displacement standard deviation of multiple threshold optimization. The stability and quality of this method are better than that of a single model. In addition, an atmospheric correction model has also been proposed to estimate the atmospheric effects on Ground-based synthetic aperture radar (GBSAR) observations. After comparing the monitoring results before and after correction, we clearly found that the results are in good agreement with the actual observations after applying the proposed atmospheric correction approach.

**Keywords:** ground-based SAR; TS30; corner reflector (CR); deformation monitoring; PS point selection; atmospheric correction

## 1. Introduction

Ground-based synthetic aperture radar (GBSAR) interferometry is a local area deformation monitoring technology that has been rapidly developed over the past two decades. Compared with traditional deformation monitoring methods such as GPS, total station, level, and laser monitoring, GBSAR has a greater observation distance, larger field of view, can penetrate rain and fog, can provide continuous space coverage and real-time monitoring, offers a high spatial resolution, and provides up to sub-millimeter measurement accuracy. Compared with spaceborne SAR, GBSAR offers a portable solution with a millimeter accuracy for displacements in a localized area [1–3]. Moreover, the whole data collection and post-processing are simple and convenient, making GBSAR an effective supplement to spaceborne SAR and conventional geodetic monitoring instruments. At present, GBSAR is primarily used to monitor landslides [2,4–6], slopes [7], Volcanic activity [8], and glaciers [9], as well as the deformation of large buildings such as dams [9–12] and towers and bridges [7,13,14]. Antonello et al. employed the InGrID-Lisa GBSAR system to monitor the Stromboli Volcano and through the radar measurement it has been possible to assess the deformation field over a large portion of the target area and to differentiate different processes [15]. Noferini et al. used a GBSAR system to monitor a landslide in northeastern Italy and proved that such a system can be used for glacial displacement monitoring [2]. Del Ventisette et al. used GBSAR to monitor the ruin on landslide in Italy for a period of one year and

concluded that there was a close correlation between surface deformation and rainfall. This showed that such a system could act as an early warning for landslide disasters [16]. Serrano-Juan et al. used GBSAR in the construction of the Lazaglerra Railway Station in Brazil [17]. Inferred GBSAR has a high sensitivity to small displacements and can accurately acquire two-dimensional deformation fields, which can help in understanding the mechanisms underlying control structure deformations and quickly determine a vulnerable deformation disaster area. In China, Yang et al. compared GBSAR observations to conventional measurement techniques and empirically validated the effectiveness and feasibility of using GBSAR to monitor open pit mine slopes [7]. Xu et al. demonstrated the feasibility to use the IBIS-S system to monitor bridge stability [13].

Regarding the selection of the persistent scatter (PS) point, Ferretti et al. defined the ratio between the standard deviation of the amplitude information and the mean of the amplitude information of each pixel in the time series as the amplitude dispersion, and select the PS point according to the amplitude dispersion information of each pixel point [18]. Kampes et al. selected PS points using the amplitude information of the pixel in the SAR image [19]. Adam tried to estimate the signal-to-noise ratio of the target points first, and then used this information to select the PS point, but this method was not effective for some scenes [20]. Hooper et al. proposed a method for selecting PS points using target point phase information by using the spatial correlation features of the interference phase [21]. The principle of GBSAR deformation interferometry is essentially identical to that of Spaceborne SAR [22]. Hence, the approach to selecting PS points for both GBSAR and Spaceborne SAR should be similar [23]. On that basis, this work proposes a joint PS points selection solution based on the average coherence coefficient, amplitude deviation index, estimated signal-to-noise ratio, and displacement standard deviation optimization. To confirm the validity of this approach, the influence of atmospheric effects over water area on bridge deformation monitoring results are experimentally accounted for and corrected.

## 2. Problem Statement and Basic Theory

### 2.1. Principle of GBSAR for Deformation Monitoring

GBSAR is an active microwave imaging radar, which obtains high-resolution two-dimensional images through the combination of linear frequency modulated continuous wave (FM-CW) technology and SAR processing technique. Using radar interference, the phase and amplitude information of the target back scattered signal can be obtained from the coherent radar image.

GBSAR usually observes the target area at a fixed location with a zero spatial baseline. The differential phase $\varphi_m$ of each pixel can thus be decomposed into

$$\varphi_m = \varphi_{disp} + \varphi_{atm} + \varphi_{noise} \tag{1}$$

where $\varphi_{disp}$ represents the phase change in the radar line of sight (LOS), $\varphi_{atm}$ is atmospheric phase delay, and $\varphi_{noise}$ is noise.

The observed phase change in the radar LOS can be expressed as

$$\varphi_{disp} = \frac{4\pi}{\lambda}d = \frac{4\pi}{\lambda} \cdot \boldsymbol{d}_{xyz} \cdot \boldsymbol{s} \tag{2}$$

where $d$ is the displacement in the radar LOS, $\boldsymbol{d}_{xyz}$ is the three-dimensional displacement, and $\boldsymbol{s}$ is the unit vector in the radar LOS.

### 2.2. PS Point Selection Rules

Most of the commonly used PS points selection solutions, such as the amplitude deviation threshold method [18], the coherence coefficient threshold method [20], and the phase deviation threshold method [24] are limited in that they usually only consider one of the characteristics of

the PS points. In order to set the thresholds of both the average coherence coefficient and the amplitude deviation index, one possible way is combining the phase deviation, the average coherence coefficient and the amplitude deviation methods and so on. Such that the initial PS points are screened through the mathematical methods (intersection or union) process [24], and then PS points stability is considered [21]. The persistent scatter control (PSC) is further selected from the primary PS points by estimating the signal-to-noise ratio and the displacement standard deviation for construction of network solution. The initial phase ambiguity estimation, the atmospheric influence, and the displacement solution are all determined spatially over time before all of the PS points are integrated into a solution for the overall deformation field [25].

In this paper, we study the average coherence coefficient and amplitude deviation threshold integration method to calculate the temporal coherence coefficient $r$ of each pixel in the image and the average coherence coefficient $\overline{\gamma}$ of each pixel in the time series. The appropriate coherence coefficient threshold can then be set to eliminate all PS points with coherence coefficient values that are less than the threshold. At the same time, the mean value $m_A$ and the standard deviation $\sigma_A$ of each pixel amplitude in the time series are calculated, and the amplitude deviation index $D_A = m_A/\sigma_A$ of each pixel is obtained. Testing several times according to actual conditions by setting a reasonable threshold value $T_d$, a cell with $D_A \leq T_d$ is initially selected as the primary PS points.

From the primary PS points, other PS points are further selected using an estimated signal to noise ratio (SNR) threshold. The estimated signal-to-noise ratio in a GBSAR system can be calculated using the average amplitude of each pixel and the standard deviation of the amplitude, expressed as

$$SNR_A = \frac{m_A^2}{2\sigma_A^2} \tag{3}$$

where $m_A$ represents the average amplitude value of each pixel in the actual design size window in the images and $\sigma_A$ represents the standard deviation of the amplitude. For the selected PS points, $SNR_A \geq SNR$ points are further selected by setting the estimated SNR threshold. Although the estimated signal-to-noise ratio threshold is higher, the selected points quality is better, but considering the density of the PS points, the threshold setting should not be too high, which should be determined according to the actual situation on site.

The Ku-band radar electromagnetic wavelength used by GBSAR is 17.4 mm. If the standard deviation of phase is $\sigma_{\varphi disp} = 20°$, the displacement standard deviation of monitoring is 0.5 mm. By setting the displacement standard deviation threshold $\sigma_d$, points less than the threshold are selected as PS points, and the specific displacement standard deviation expression can be expressed as

$$\sigma_d = \frac{\lambda}{4\pi}\sigma_{\varphi disp} \tag{4}$$

where $\sigma_d$ is the standard deviation of displacement and $\sigma_{\varphi disp}$ is the standard deviation of phase.

Although appropriately setting the PS point selection threshold improves the quality of the PS points, the PS point density needs to be sufficient to ensure continuous data processing after interpolation. Thus, the threshold cannot be set too high and should be determined empirically based on the characteristics of the area being observed.

## 3. Case Studies

### 3.1. Case Study 1: Yuehu Bridge Movement Monitoring

The Yuehu Bridge used in Case Study 1 is located at Yuehu District, Xiangtan City, Hunan, China. It is a landscape bridge across the artificial lake. The bridge spans about 70 m and is 11 m wide. The bridge mainly uses pedestrians and non-motor vehicles. The Yuehu Bridge is a reinforced concrete structure located on the lake. No motor vehicle passed over the bridge during the experiment and there was a clear LOS of GBSAR. That is, the emitted radar signal covered the entire Yuehu

Bridge without any obstruction between the radar system and the bridge (Figure 1). In order to assess the performance of GBSAR, a Leica TS30 total station (nominal distance standard uncertainly: 0.6 mm + 1 ppm) collected measurements prism position data simultaneously. Five corner reflectors were fixed on the bridge with a prism being placed near to each CR, as shown in Figure 1. GBSAR and Leica TS30 total station is about 170 m away from Yuehu Bridge. The basic parameters of GBSAR during the detection are set as: Ku band, 17.2 GHz; bandwidth is 199.9 MHz; pulse repetition frequency is 375.1 Hz; range resolution is 0.5 m; azimuth resolution is 4.5 mrad; scan interval is 30 s. The Leica TS30 total station measurement interval is 10 min each time. Finally, the experiment collected 30 sets of Leica TS30 total station data and and monitored 350 GBSAR image scenes.

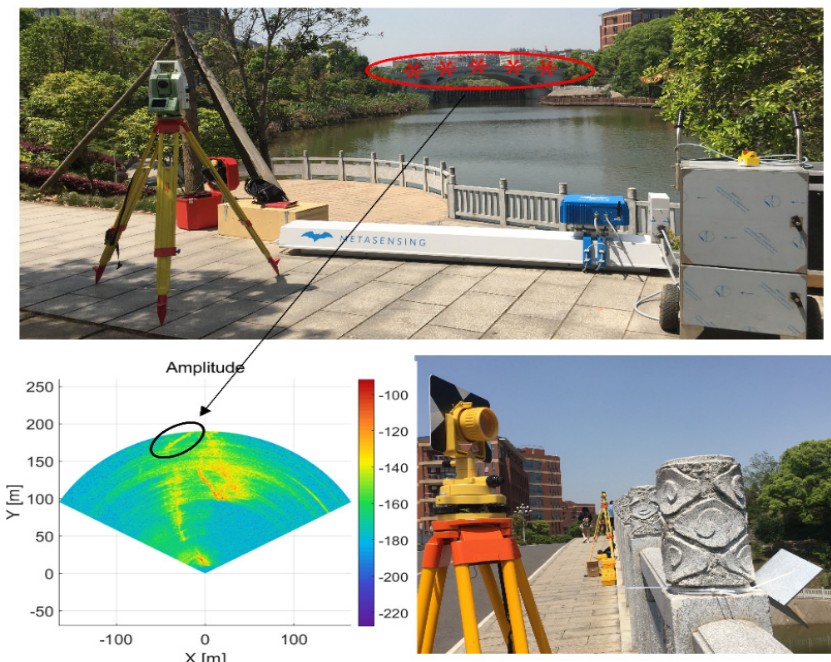

**Figure 1.** Experiment site of the Yuehu Bridge and associated electromagnetic wave reflection energy diagram.

### 3.1.1. PS Point Detection

In order to reduce clutter interference from other ground objects outside the bridge area, PS points were selected using the new solution described in Section 2.2. The PS points were selected based on the amplitude, deviation threshold, and coherence coefficient thresholds, and only the Yuehu Bridge was used for further data processing and analysis. The filter window was set to reduce the impacts of noise and ensure the continuity of the phase so as to avoid phase unwrapping errors.

Error sources for GBSAR include observation platform instability, atmospheric delay, system frequency deviation errors, and interference phase errors and so on, among which atmospheric delays represent the largest error [26]. Using the combined optimal selection solution proposed in this work, PS points from the 350 scene images were extracted and collected. After several tests, different thresholds were set for each of the four parameters: an amplitude deviation threshold of 0.4, an average coherence coefficient threshold of 0.9, an estimated signal-to-noise threshold value of 15, and a standard deviation of displacement of 0.4. A total of 538 PS points was obtained after the intersection extraction, as shown in Figure 2, in which the x-axis represents the lateral distance for each PS point to the monitoring instrument and the y-axis represents the longitudinal distance between each PS point and the monitoring instrument. Since the actual area occupied by the bridge in the extracted image was small, the size of the PSC search grid was set to $5 \times 5$ m$^2$ and the threshold was set to 0.2. As shown in Figure 3, 12 PSC points were obtained for the bridge.

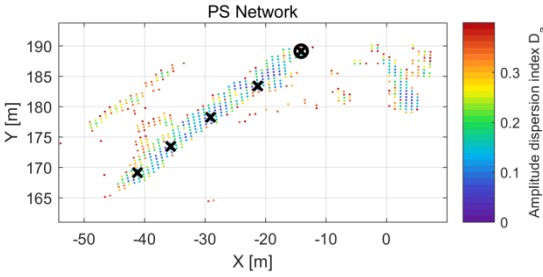

**Figure 2.** Result of selected PS points.

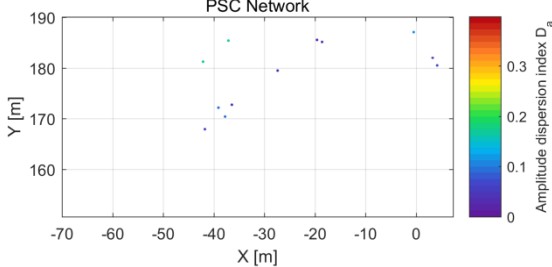

**Figure 3.** Result of selected PSC representative points.

After image collection, the average coherence coefficient of each point was extracted, along with the amplitude deviation index, the estimated signal to noise ratio, and the displacement standard deviation. As shown in Table 1, each PSC point parameter is better than the experimental set value, and the amplitude dispersion index is less than 0.2, indicating that the scattering characteristics of each PSC point obtained are relatively stable. The coherence of each point is also at least 0.917, and the mean value is about 0.95, which indicates that the correlation coefficient of these points in the PS point set in the time series changes little with time. The signal-to-noise ratio is positively correlated with the coherence coefficient, and most of them are above 25 dB, which means that the measurement process is less affected by the external environment and the instrument itself. The difference in the standard deviation of displacement in the PSC point set is relatively large, which is related to the change of the area of each PSC point in the measurement process. One of the maximum displacement standard deviations reaches 0.431 mm. The analysis based on the area of this point may be due to the process of measurement, it is affected by the influence of pedestrians and vehicles, but it does not affect the deformation analysis results of the entire area. Overall, the four parameters of each PSC feature point reach the thresholds which is necessary to meet the extracted requirements of PS points.

**Table 1.** PSC parameters.

| X(m) | Y(m) | Amplitude Deviation Index | Average Coherence Coefficient | Signal to Noise Ratio (SNR) Estimate | Standard Deviation of Displacement |
|---|---|---|---|---|---|
| −42.1 | 181.3 | 0.167 | 0.935 | 24.35 | −0.184 |
| −41.8 | 187.1 | 0.124 | 0.950 | 23.69 | −0.195 |
| −39.1 | 172.2 | 0.091 | 0.994 | 24.82 | 0.091 |
| −37.8 | 170.4 | 0.088 | 0.928 | 23.19 | 0.268 |
| −37.12 | 185.42 | 0.185 | 0.917 | 19.54 | −0.180 |
| −36.4 | 172.8 | 0.037 | 0.989 | 26.55 | −0.083 |
| −27.4 | 179.5 | 0.037 | 0.992 | 27.68 | 0.431 |
| −19.6 | 185.6 | 0.034 | 0.987 | 26.31 | 0.085 |
| −18.6 | 185.2 | 0.038 | 1 | 27.86 | 0.065 |
| −0.5 | 187.1 | 0.124 | 0.950 | 23.69 | −0.195 |
| 3.3 | 182.1 | 0.045 | 1 | 29.89 | −0.088 |
| 4.2 | 180.5 | 0.070 | 0.983 | 25.89 | −0.079 |

### 3.1.2. Atmospheric Effects

The Leica TS30 total station and GBSAR observed the bridge simultaneously. TS30 collected one measurement every 10 min. It acquired 28 sets of distance measurements between TS30 and the five prisms. The offset of the five viewing prism positions is shown in Figure 4. Since the range of variation is in the range of 1 mm, the chart retains only its mm value. Each data of the results show that the distance fluctuation is within the range of the instrument nominal standard deviation of 0.6 mm, indicating that there was no deformation of the bridge.

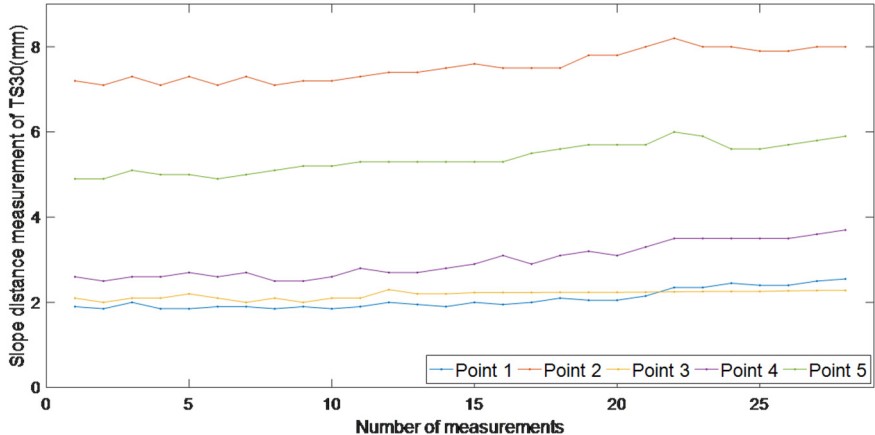

**Figure 4.** The fluctuation diagram of the slope distance measurements of TS30.

For the five Corner Reflectors, PS points were extracted from the GBSAR images. The time series changing curves of deformations are shown in Figure 5. It can be clearly seen that the deformation time series curve of each point shows a tilt change trend, and the maximum point position has a variation of nearly 2 mm, but actually the CR and the above monitored prism position were stationary, and there was no deformation, indicating that the radar measurement exhibited atmospheric delays. The meaning of **cuw** in the Figure 5 is Coordinate UnWrapped.

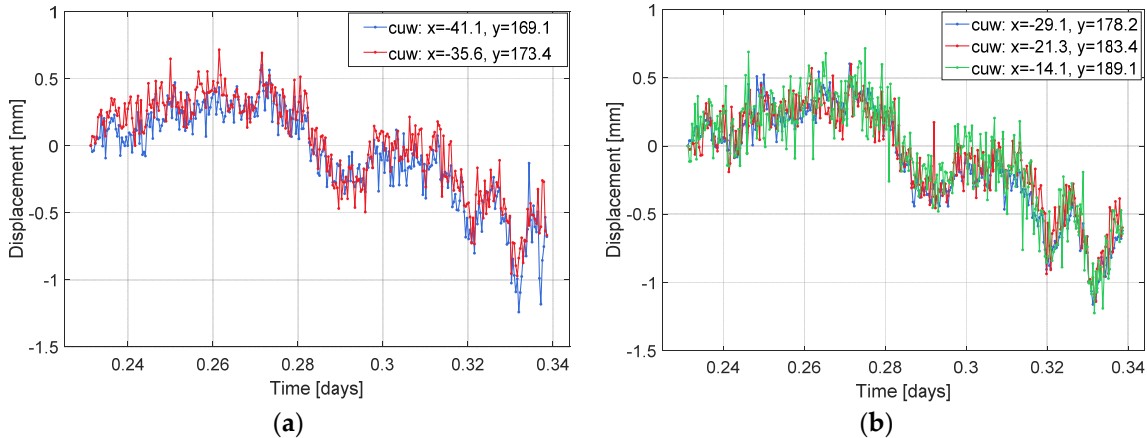

**Figure 5.** LOS displacement before atmospheric correction of the five representative PSs. (**a**) indicates points (−41.1, 169.1) and (−35.6, 173.4); (**b**) indicates points (−29.1, 178.2), (−21.3, 183.4) and (−14.1, 189.1).

Because of the short distance observations, no sudden change of weather occurred during the observation period, and we assume that the experimental area was in a uniform atmospheric environment and atmospheric effects were only related to the radar travel path. Thus, the relationship can be expressed as:

$$\varphi_{atm} = \frac{4\pi}{\lambda}ar \tag{5}$$

where $a = 7.76 \times 10^{-5}\frac{P}{T} + 3.73 \times 10^{-1}\frac{e}{T^2}$ [27] and $r$ is the distance between the sensor and the target. The 12 PSC points were then used to establish the atmospheric correction model:

$$\varphi_{atm} = \frac{4\pi}{\lambda}\sum_{i=0}^{m} a_i r^i \tag{6}$$

where $m$ is the number of polynomials.

Through setting different values of $i$, according to the number of calibrations in Equation (6), we found that the best value for the polynomial number m to fit the data was 2. Figure 6 shows the results for the five PS points of the atmospheric phase diagram. It can be seen that the atmosphere phase has significant influence on the measurement results. According to the graph of atmospheric phase curve, as the monitoring time extends the generated atmospheric impact also adds.

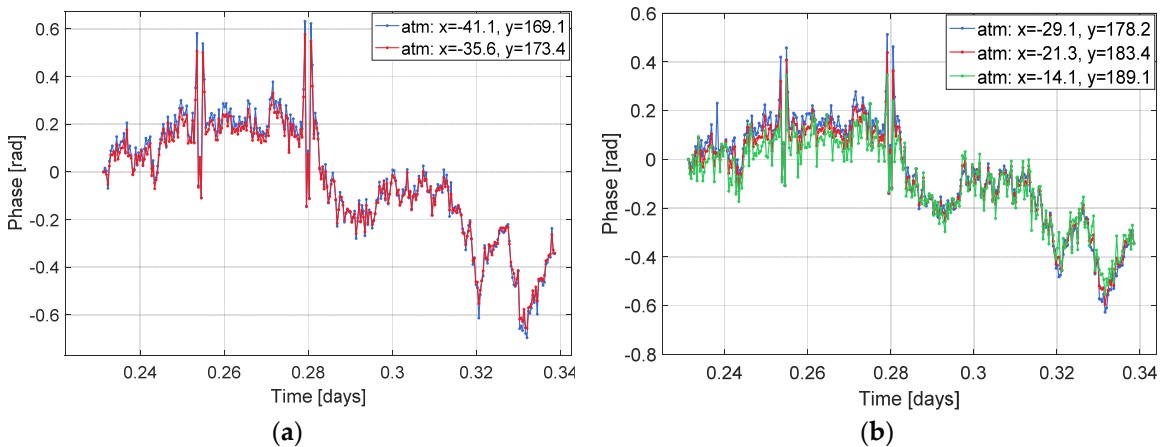

**Figure 6.** Atmospheric phase changes of five representative PS points. (**a**) indicates points ($-41.1$, 169.1) and ($-35.6$, 173.4); (**b**) indicates points ($-29.1$, 178.2), ($-21.3$, 183.4) and ($-14.1$, 189.1).

### 3.1.3. Results

Using the atmospheric correction model from the previous section, all the radar observations were corrected, as shown in Figure 7. A comparison of the deformation curves before (Figure 5) and after (Figure 7) atmospheric correction clearly illustrates that the deformation values of the target points selected in the monitored time period are mostly $-0.5$ to 0.5 mm in the sight direction of the radar. That is, the changes in displacement are within 1 mm.

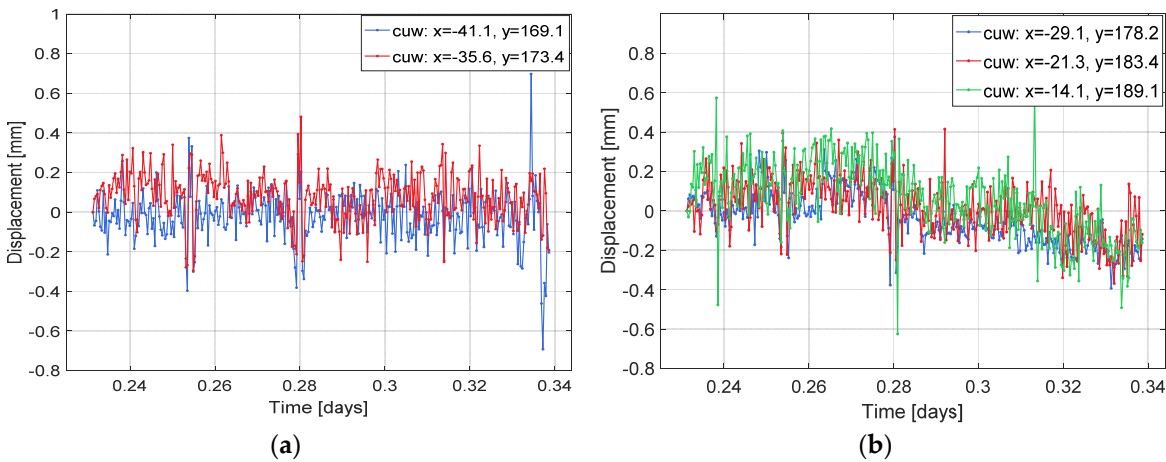

**Figure 7.** LOS displacement after atmospheric correction of the five representative PS points. (**a**) indicates points ($-41.1$, 169.1) and ($-35.6$, 173.4); (**b**) indicates points ($-29.1$, 178.2), ($-21.3$, 183.4) and ($-14.1$, 189.1).

Based on the monitoring results of the five PS points, the LOS deformation mean value and root mean square (RMS) error before and after atmospheric correction at each point were calculated, and the LOS direction deformation value comparison chart before and after atmospheric correction was obtained, as shown in Table 2. As can be seen from the figure, the average deformation before correction is larger than that after correction, and the maximum correction for atmospheric can reach 0.25 mm. At the same time, the RMS error of deformation after correction is smaller than that of deformation before correction. It can be seen that the monitoring results after atmospheric correction are closer to the measured values.

**Table 2.** Comparison of average displacements of five PS points before and after atmospheric correction.

| Point | Before Correction Average Deformation Value (mm) | After Correction Average Deformation Value (mm) |
|-------|--------------------------------------------------|-------------------------------------------------|
| P1 | 0.3 | 0.05 |
| P2 | 0.29 | 0.17 |
| P3 | 0.34 | 0.2 |
| P4 | 0.29 | 0.09 |
| P5 | 0.38 | 0.18 |

All the PS points are applied joint interpolation calculation before and after atmospheric correction, and the deformation results of the Yuehu bridge were shown as Figure 8. Comparison of the two graphs shows that, before correction, the deformation was −4 to 3 mm, while the deformation after atmospheric correction was only −1 to 1 mm, indicating that the PS point atmospheric correction solution can reduce atmospheric delays.

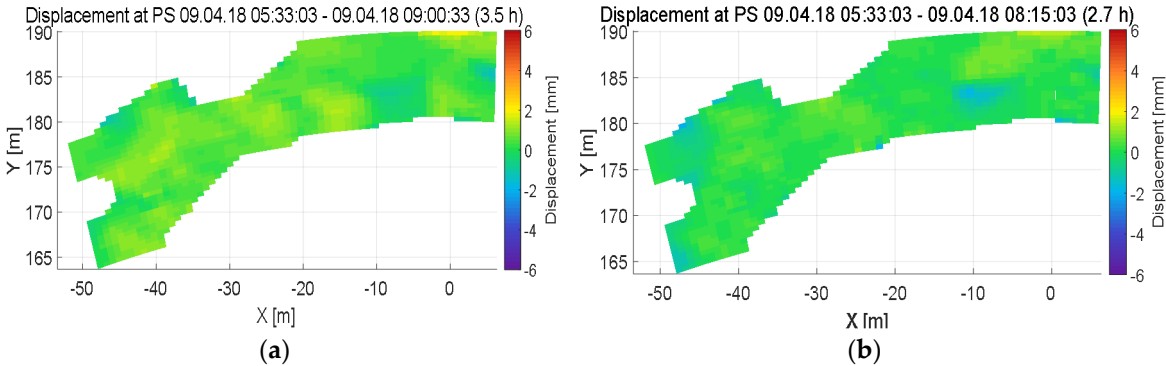

**Figure 8.** Deformation of the Yuehu bridge before and after atmospheric correction. (**a**) indicates before atmospheric correction; (**b**) indicates after atmospheric correction.

For monitoring areas with short distances, the joint PS point atmospheric correction solution can reduce the influence of atmospheric delay and improve deformation monitoring accuracy.

*3.2. Case Study 2: Earth Wall Scarp Deformation Monitoring*

The small earth wall slope used in Case Study 2 is located at Yuehu District, Xiangtan City, Hunan, China. It is about 5 m high and 40 m wide. The terrain is relatively flat in front of the slope, predominantly vegetable fields, with low vegetation cover on the left side. A cement concrete foundation platform with good visibility about 70 m away from the earth wall slope was selected as the GBSAR instrument observation position, due to its stability, and four corner reflectors were placed on the slope (Figure 9). The test lasted about 5 h, and a total of 510 radar images were collected.

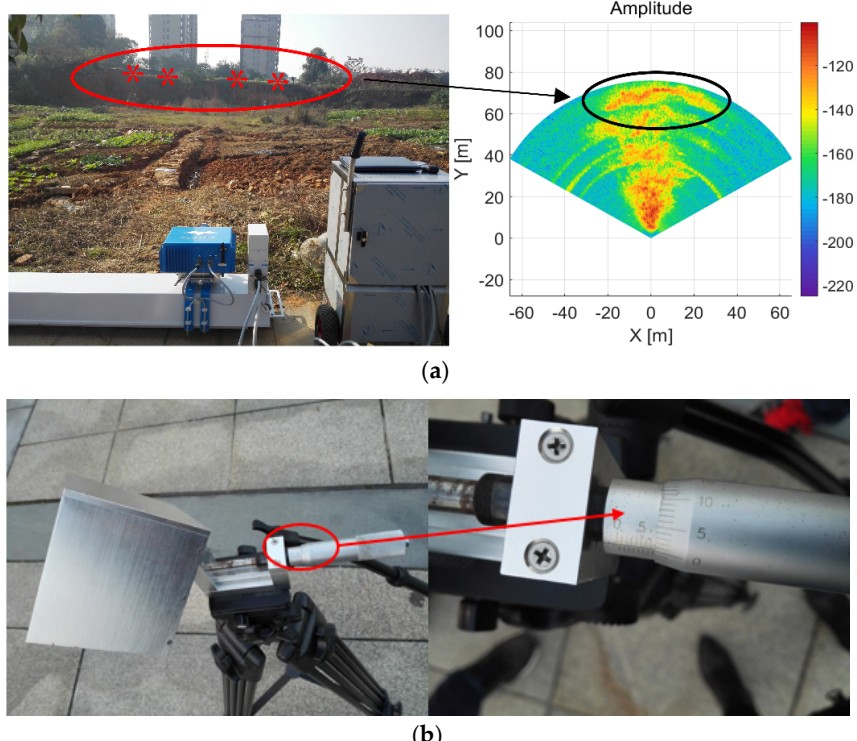

(a)

(b)

**Figure 9.** Electromagnetic wave reflection energy diagram experiment site of the small slope with adjustable CR. The picture on the left of part (**a**) shows the scene of the small earth wall slope monitoring. The positions of the four corner reflectors are marked in the figure, and the right picture shows the related electromagnetic reflection energy map. (**b**) is the adjustable corner reflector with measuring ruler.

### 3.2.1. PS Point Detection

PS point extraction was performed on the image. After experimental analysis, the PS point amplitude deviation threshold was set to 0.4, the average coherence coefficient threshold was 0.7, the estimated SNR threshold was 10, and the standard deviation threshold was 0.4. After the intersection extraction from the thresholds, a total of 801 PS points was obtained. As shown in Figure 10, where the X-axis represents the lateral distance of each PS point from the monitoring instrument, and the Y-axis represents the longitudinal distance of each PS point from the monitoring instrument. The size of the PSC points search grid area was also set to $5 \times 5$ m$^2$, and the threshold was set to 0.2, then a total of five PSC representative points were obtained, as shown in Figure 11.

The average coherence coefficient, amplitude deviation index, estimated signal-to-noise ratio, and standard deviation of displacement of each point of PSC were extracted. As shown in Table 3, the four parameters of each PSC feature point are within the threshold range to meet the selection requirements of PS points.

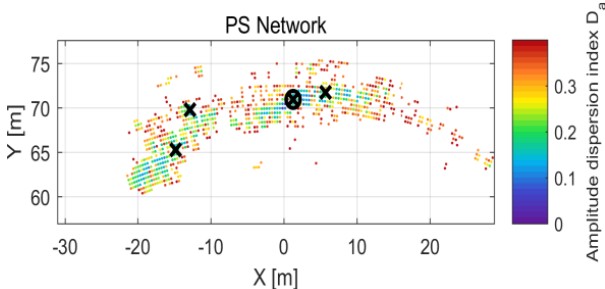

**Figure 10.** Result of selected PS points.

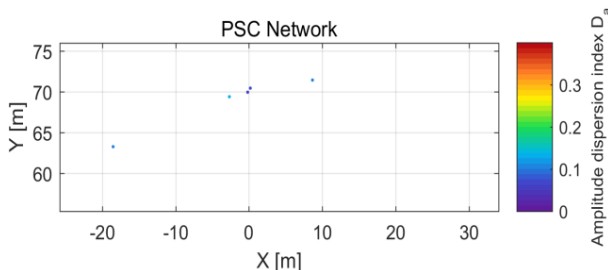

**Figure 11.** Result of selected PSC representative points.

**Table 3.** Four parameters of PSC.

| X(m) | Y(m) | Amplitude Deviation Index | Average Coherence Coefficient | SNR | Standard Deviation of Displacement |
|---|---|---|---|---|---|
| −18.52 | 63.28 | 0.101 | 0.965 | 24.35 | 0.017 |
| −2.67 | 69.41 | 0.134 | 0.932 | 23.19 | −0.193 |
| −0.18 | 69.96 | 0.053 | 1 | 29.33 | 0.034 |
| 0.18 | 70.47 | 0.069 | 1 | 28.45 | 0.209 |
| 8.66 | 71.45 | 0.101 | 0.985 | 25.87 | −0.067 |

### 3.2.2. Atmospheric Effects

To verify the accuracy of the test, an adjustable CR with measuring ruler was adjusted five times during the monitoring period, and set to 3 mm, 4 mm, 3 mm, 3 mm, and 3 mm, respectively. Four corner reflectors PS points (Figure 10) were extracted from the GBSAR images for verification. The time series curves of the deformation result are shown in Figure 12. It can be clearly seen from Figure 12 that the manual adjustment displacement of the adjustable CR with measuring ruler is clearly identified, as shown in Figure 12a,b. At the same time, the time curves of the four corner reflectors show a large tendency of system tilt change. The largest point has a variation of approximately 2 mm. During the monitoring period, the monitoring area was stable, and the CR was a fixed device on the earth wall slope, there was no deformation. It can be seen that the measurement results have the influence of noise errors such as atmospheric delay.

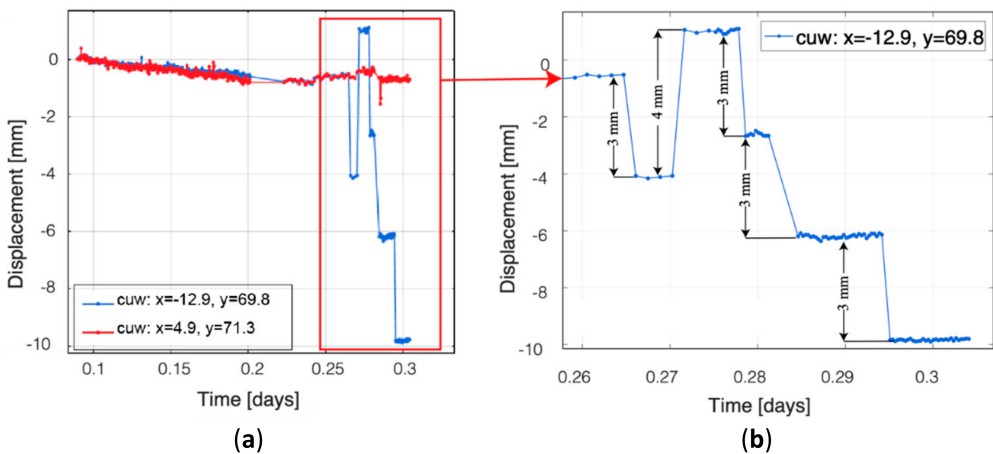

**Figure 12.** *Cont.*

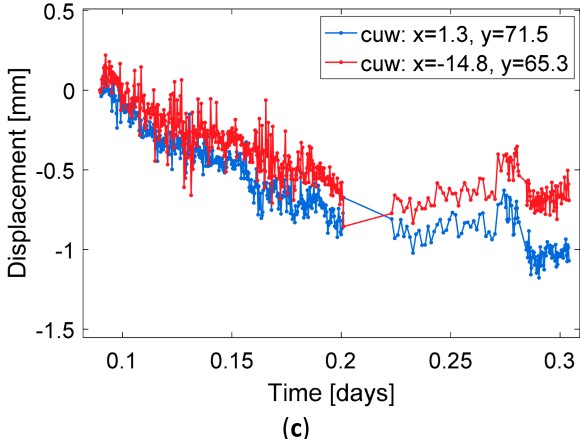

**(c)**

**Figure 12.** Displacement of sight line before atmospheric correction of the four representative PSs. (**a**) indicates points (−12.9, 69.8) and (4.9, 71.3); (**b**) is the box portion in (**a**), the corner reflector adjustment size has been marked in the figure, and (**c**) indicates the point (1.3, 71.5) with (−14.8, 65.3).

As shown in Figure 13, atmospheric delay for four PS points has been derived according to the atmospheric delay estimation method mentioned above. It can be clearly found out that the atmosphere delay has a great impact on the results, with maximum value reached 2 mm. We also find that with the longer amount of monitoring time, the cumulative impact of atmospheric effects will increase.

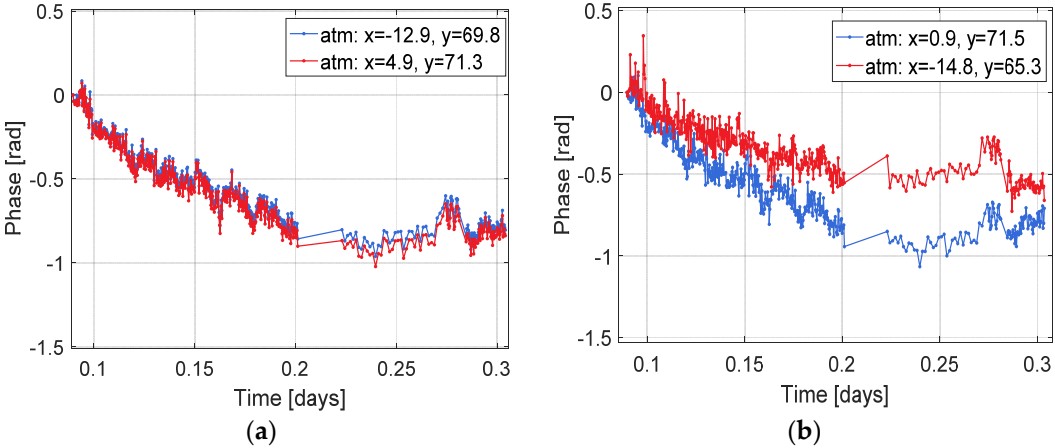

**(a)**　　　　　　　　　　　　　　　　　**(b)**

**Figure 13.** Atmospheric phase of the four representative PSs. (**a**) indicates points (−12.9, 69.8) and (4.9, 71.3); (**b**) indicates the point (1.3, 71.5) with (−14.8, 65.3).

### 3.2.3. Results

The atmospheric phase delay correction was performed on each PS point to obtain the deformation result after correction, as shown in Figure 14. Comparing the deformation curves before and after the atmospheric phase delay correction in Figures 12 and 14, it can be clearly seen that after the atmospheric phase delay correction, the target point selected in the monitoring period is between −0.5 mm and 0.5 mm in the radar sight line, which is the displacement change value. It fluctuates within 0.8 mm. At the same time, the deformation values monitored by the adjustable CR with measuring ruler are 2.8 mm, 4 mm, 3.1 mm, 3 mm and 3.2 mm respectively. The monitoring results are very close to the results of the manual set change of measuring ruler, suggesting that this solution of atmospheric delay correction is effective.

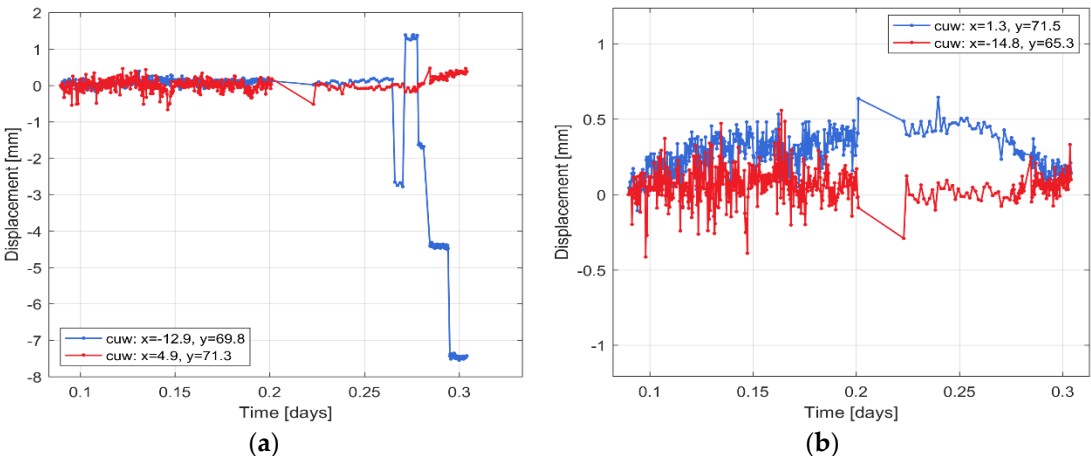

**Figure 14.** Displacement of sight line after atmospheric correction of the four representative PSs. (**a**) indicates points (−12.9, 69.8) and (4.9, 71.3); (**b**) indicates the point (1.3, 71.5) with (−14.8, 65.3).

The fusion processing results of other PS points on the earth wall slope before and after atmospheric phase delay correction are shown in Figure 15. It can be found that the deformation before correction is between −3 to 2 mm, and the deformation after correction is about −1 to 1 mm, and the actual deformation of the monitoring area shows that the PS points correction solution can weaken the effect of systematic errors such as atmospheric phase delay and improve the deformation monitoring accuracy.

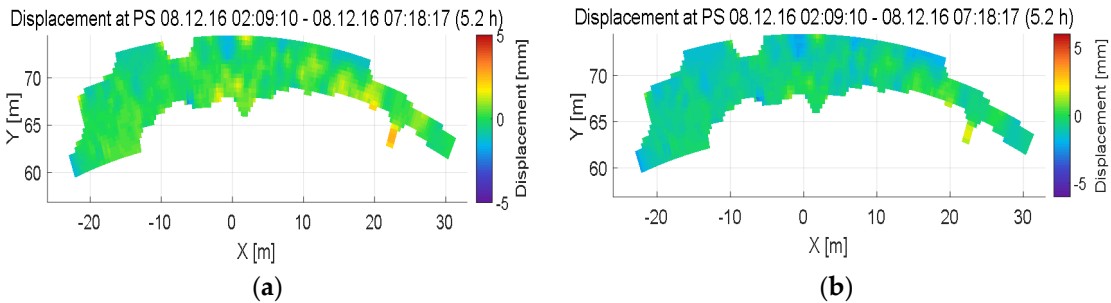

**Figure 15.** Deformation results before and after atmospheric correct. (**a**) indicates before atmospheric correction, and (**b**) indicates after atmospheric correction.

## 4. Discussion

In this paper, the selection methods of GBSAR PS points and the establishment of atmospheric correction model of PS points triangular network were put forward, in which the deformation of Yuehu Bridge and the small slope were monitored respectively, taking the atmospheric influence into consideration. Based on the above experimental methods and results, four questions need to be discussed. Details are as follows.

(1) In Yuehu Bridge monitoring tests, the selection of PS points was analyzed, and atmospheric effect correction model was built. The adoption of PS points of atmospheric correction methods can effectively improve the monitoring precision by reducing atmospheric effects on the condition that the monitoring of the surrounding environment is stable, and the target is in a short-range monitoring area. However, the atmosphere effect correction model needs further improvement for the complex terrain, weather, unstable regions of the application.

(2) In the small earth wall slope monitoring tests, the adjustable CR with measuring ruler arranged are inadequate in number due to the limited test conditions. Other PS points are manually selected as a complement to build the triangulation atmospheric correction model, which can, to a certain extent, improve the precision of the monitoring results. But the influence of the atmosphere cannot

be completely eliminated, so there still exists some residual error. Therefore, more experiments are needed to test the effect of the atmospheric correction network with more CR reflectors.

(3) In order to reduce the interference of clutter signals, only the image data of the Yuehu Bridge was selected for data processing. In the process of PS points selection, points that are less than the threshold are selected by setting the average correlation coefficient and amplitude deviation threshold integration methodology. The threshold should not be set too high, but set according to the actual situation, because the density of PS points needs to be considered from the perspective of data processing, although the threshold settings are adopted to improve the quality of PS points.

(4) Errors affecting GBSAR include systematic frequency deviation error, observation platform instability error, and atmospheric delay effect, etc. In this paper, the interference of atmospheric effects on radar echo signal was studied. The monitoring experiment of the Yuehu Bridge and the small earth wall slope showed that selecting the appropriate atmospheric correction model can effectively improve the precision of GBSAR monitoring results. Under the condition of long-time observation, the effect of platform instability, etc. should also be considered.

## 5. Conclusions

In this paper, a PS Optimization Selection Method was put forward that takes into account the average coherence coefficient, the amplitude deviation index, the estimated signal to noise ratio, and the displacement accuracy index. This method effectively improved the quality of the selected PS points. In addition, atmospheric delay correction method was proposed, in which the monitoring experiment of Yuehu Bridge and the small earth wall slope is carried out respectively. After comparing the Leica TS30 total station and the adjustable CR with measuring ruler, it is shown that the GBSAR system could achieve sub-millimeter high precision monitoring after atmospheric correction for short-range deformation monitoring. More specifically, the results presented in the paper clearly highlight that:

(1) In the Yuehu Bridge deformation monitoring test, the GBSAR system and the Leica TS30 total station were used for comparative analysis. Simultaneous monitoring of the same target was performed. After data processing and analysis, the monitoring results of the GBSAR system and the Leica TS30 total station values were identical with each other and consistent with the actual situation on site.

(2) After expounding the conventional PS points selection methods, we put forward a GBSAR PS points selection method based on the average coherence coefficient, amplitude dispersion index, estimated signal-to-noise ratio and displacement accuracy index, and so on. An empirical study and analysis of Yuehu Bridge and small slope were carried out, and the PS points with higher quality than the single method were successfully selected. The atmospheric correction model constructed by PS point was used to correct the monitoring results of the Yuehu Bridge and the small earth wall slope. The comparative analysis of the deformation time series curves before and after the atmospheric correction shows that the use of PS points to construct the triangulation network for atmospheric correction has achieved good results.

(3) In the small slope monitoring test, the collected data were processed by focusing, interference, unwrapping phase, etc., and the final displacement map was obtained. According to the actual situation analysis, it was concluded that the atmospheric effect is the main factor affecting the precision of the test results. By comparing the adjustment of the CR with measuring ruler and the GBSAR monitoring results, it was concluded that the GBSAR has sub-millimeter-level monitoring precision after atmospheric correction.

**Author Contributions:** S.L. conducted the algorithm design. S.L., Y.Y. and A.T. wrote the paper. S.L. and Z.L. revised the paper. Formal Analysis, S.L., Y.Y. and A.T.; Data Curation, Y.Y. and A.T.; Investigation, Y.Y., A.T., W.W., C.Z.; Resources, Y.Y. and A.T.; Supervision, S.L. and Z.L.; Project Administration, S.L.; Funding Acquisition, S.L. All authors have contributed significantly and have participated sufficiently to take responsibility for this research.

**Funding:** This work was supported by the National Natural Science Foundation of China (grant 41877283, 41474014), and the Study Abroad Fund of the Hunan Provincial Department of Education (2015212), and the Scientific Research Fund of Hunan Provincial Education Department (15A060).

**Acknowledgments:** Thanks to Fast-GBSAR system for providing accurate test data, and thanked the company's technical staff Li Chen and Zheng Wang in Newcastle University for their technical support.

**Conflicts of Interest:** The authors declare no conflict of interest.

## Abbreviations

The following abbreviations are used in this manuscript:

| | |
|---|---|
| CR | corner reflector |
| FM-CW | Frequency Modulated Continuous Wave |
| GBSAR | Ground-Based Synthetic Aperture Radar |
| GPS | Global Positioning System |
| InSAR | Interferometric Synthetic Aperture Radar |
| LOS | Line of sight |
| PS | Permanent Scatterers |
| PSC | Permanent Scatterers Candidates |
| RMS | Root Mean Square |
| SNR | Signal to Noise Ratio |

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
