# Peer review of "New Approaches to Processing Ground-Based SAR (GBSAR) Data for Deformation Monitoring"

_remotesensing, doi:10.3390/rs10121936_

Round 1
Reviewer 1 Report
In this paper, the authors applied PSI analysis for GBSAR system. The proposal is interesting and the results are reasonable. However, multiple modifications are required prior to be published.
1. First of all, PSI method is useful for long term time-series measurement. GBSAR system has short temporal baseline that almost all points have high coherence. To show the applicability of the proposal, it is required to compare the results with a simple stack of the interferograms. At the same time, there is no truth data to confirm the accuracy of the results.
2. What is the “traditional deformation measurement technology” in L. 27? For example, laser monitoring can measure the target with sub-millimeter accuracy with more frequent observation than GBSAR. In the following sentence, the authors compared spaceborne SAR and GBSAR. Do the same thing to the first one.
3. Please make it clear that from where the authors' proposal starts. Probably it is L. 94 but currently it is not clear.
4. In section 3, the target is not familiar with non-Chinese people. Please provide more information such as location, size and so on.
5. Please provide more specific information for GBSAR and TS30 robot. E.g., bandwidth, observation length, revisits cycle and so on.
6. In Section 3.1, please provide where the five representative PSs are in photography.
7. Please cite a specific reference for the parameter "a" in (5).
8. Rectangular in the left top of Fig. 12 (please label them (a), (b) and (c)) is not matched with the right top one. The left border must be close to 0.26 and the right must exceed 0.3.
9. Some sentences are too long and not easy to understand. For example, L. 12-16 and L. 85-90 use 5-6 lines for single sentence. Please revise the manuscript to simplify them.
10. What are “cuw” in the graphs?
11. Supplementary Materials, Author Contributions and Acknowledgments are not written.
12. The authors did not use [22].
Author Response
New approaches to processing ground-based SAR (GBSAR) data for deformation monitoring
Manuscript No: remotesensing-388171
Sichun LONG, Aixia TONG, Ying YUAN, Zhenhong LI, Wenhao WU, Chuanguang ZHU
---
Dear,
Many thanks for the constructive and encouraging comments on our manuscript from yourself.We enclose a carefully revised manuscript and an item-by-item response according to the comments and suggestions made. The comments are included in “bold italics”, and our responses are in “regular” text.
We hope that these clarifications and revisions will now enable the paper to be accepted for publication in Journal of Remote Sensing, and look forward to hearing from you soon.
Yours sincerely,
Professor Sichun Long (on behalf of all authors)
Hunan University of Science and Technology, Xiangtan Hunan, China, 411201;

Reviewer 2 Report
Dear authors,
I reviewed the paper entitled “New approaches to processing ground-based SAR (GBSAR) data for deformation monitoring” by Sichun Long et al. The paper shows an interesting method for selecting Persistent Scatterers that combines different indexes to improve their quality. Then, two experiments are exposed to show its applicability limited to short-range distances after APS correction.
The paper is not well written and should be revised by a native English speaker. It is difficult to understand in some parts and there are many flaws and spelling mistakes. Related to the structure of the paper, the introduction offers enough literature review. Note that the reference [22] is not cited in the text. The methodology section is brief but includes the necessary information. The experiments are correctly described although I will detail some indications below. In the conclusions section, more information about the obtained results should be given to conclude about the strengths of the proposed method. The discussion section is mixed with the results, so maybe it would be better to separate them. First, describe your experiments and results and after that, include a discussion section before the conclusions. In this discussion section you could discuss and analyze the benefit of your methodology comparing with some other experiments from other authors, also in short-range distances like yours, in which they use a single index for PS selection. This would enrich the paper and stress the advantages of your methodology respect to different criteria for the PS selection. The references must be revised as do not have a uniform style.
The manuscript does not have the lines correctly numbered, so it is more difficult to indicate the changes.
Find below some specific comments:
In Affiliations, the name of the country is missing.
Line 13 -> The correct word is “Scatterers”, not “Scatter”. Check this along your manuscript including “Abbreviations” section at its end part.
Line 32 -> This sentence is not understandable. Something is missing.
Line 35 -> “eroputions”.
Line 35 -> Is it correct to include Reference [2] as a dam case? It is a landslide study. What do you mean with “large buildings such a dams”?
Line 47 -> “compared” not “Compared”.
Line 50 -> Add the appropriate reference. Is it [13]?
Line 51 -> Replace “…ratio of the…” with “…ratio between the…”.
Unify “line of sight” vs. “line-of-sight” along the paper. In some other parts of the paper you use LOS. So, indicate LOS the first time you use “line-of-sight” or “line of sight” and then use LOS after that in the text.
Line 86 -> “phase” not “Phase”.
Lines 85-90 -> It is along sentence, not well understandable. Rewrite it. “though” is not correct in this sentence.
Line 96 -> Add a space between gamma and “of”.
Line 99 -> “Test several…” is not well understandable. Do you mean “Testing”?
Lines 75, 106, and 117 -> Write “where” instead “Where” at the beginning of these lines as in line 79.
Line 113 -> Add units after “0.5”.
Line 124 -> Indicate where this bridge is located.
Line 128 -> (Figure 1) instead (Figure 1.).
Line 128 -> “…a TS30 robot…” is not well expressed. It is a “total station” not a robot. So, for instance, write “… a Leica TS30 total station…”
Line 129 -> “accuracy” is not correct. This is the “standard uncertainty”. Why “Corner Reflectors” are with capital letter?
Line 129 -> “…collected measurements simultaneously…”. Specify which kind of measurements.
Line 131 -> What is “..TS30 measurement robot data…”? Express it correctly.
Lines 131 and 132 -> “were collected” is repeated. Rewrite better the sentences.
Which is the distance between the GBSAR and the bridge? It is not explained.
Photos in Figure 1 are too small. Better enlarge the figure. In the bottom right figure, the prims and tripod is tilted.
Line 153 -> Replace “square meters” with “m2”.
Line 164 -> Remove “(Decibel)”. It is said with the symbol “dB”.
Line 168 -> Add units to 0.431.
Line 170-172 -> The verb is missing in the sentence.
Table 1. Add units to X, Y and Standard deviation of displacement. Is m the unit for X and Y? In such a case do not write X/m and Y/m. Center vertically all the texts in header.
Line 175 -> Rewrite “TS30 measurement robot”. I would use the term “total station” when referring to the instrument, instead of calling “TS30”.
Line 177 -> “prims” instead “Prims”.
Line 179 -> It is not “accuracy” but “standard deviation”.
Figure 4 -> In the caption you say “slope measurements” but it should be “slope distance measurements”.
Line 184 -> Add a space in “Figure5”. Also, in line 186 “2mm” and in section 3.1.3, 0.25mm. In general, separate numbers and units in all the text.
Section 3.1.3. Use symbols for units (millimeters). Revise all the manuscript related to this point.
Table 2. Edit the text of the header appropriately.
In the paragraph below Table 2, the sentence of the first two lines is not understandable. Review this paragraph as the number and units are not grammatically well expressed. Before atmospheric correction I cannot appreciate that the deformation is between 0 and 7 mm. In such a case, we should see pixels in red color according to the color bar.
Section 3.2. Start describing the second example with an introduction, not just one sentence as if the reader were familiar with the study area or the experiment itself. As in the first example, it is not said the distance between the radar and the object.
Figure 9. Enlarge the photos. Edit the caption (Figure.9 Electromagnetic…). “with” not “With”.
Section 3.2.1. Line 4. The total number of PS is missing.
Section 3.2.2. Revise “micro metric” and write the units correctly, not after each number. The same in section 3.2.3.
Revise the paragraph below Figure 12. It is not well understandable.
In the text below Figure 14, check the sentences. “are shown” not “as shown”. As before, I cannot appreciate from Figure 15 that the deformation before atmospheric correction is between 0 and 5 mm. According to the color bar, 5 means red colors and they are not seen in the figure.
Supplementary Materials and Author Contributions are not well written.
Author Response

(The authors gave the same response as above.)

Round 2
Reviewer 1 Report
The modified article is fine. Please revise the following two points.
1. In L. 133, "Ku band 172.6 Hz" is too low. Probably it is "GHz" but please make it clear.
2. In L. 133, the bandwidth cannot be negative value. It must be the central frequency and the absolute bandwidth.
Author Response
Dear,
Many thanks for the constructive and encouraging comments on our manuscript from yourself.We enclose a carefully revised manuscript and an item-by-item response according to the comments and suggestions made. The comments are included in “bold italics”, and our responses are in “regular” text.
We hope that these clarifications and revisions will now enable the paper to be accepted for publication in Journal of Remote Sensing, and look forward to hearing from you soon.
Yours sincerely,
Professor Sichun Long (on behalf of all authors)
Hunan University of Science and Technology, Xiangtan Hunan, China, 411201;
Response to Reviewer #1
Main Comments
1. First of all, 1. In L. 133, "Ku band 172.6 Hz" is too low. Probably it is "GHz" but please make it clear.
2. In L. 133, the bandwidth cannot be negative value. It must be the central frequency and the absolute bandwidth.
Re: Thank you for your valuable suggestion. We carefully checked the instrument manual and corrected it as follows: “The basic parameters of GBSAR during the detection are set as: Ku band, 17.2 GHz; bandwidth is 199.9 MHz; pulse repetition frequency is 375.1Hz; range resolution is 0.5 m; azimuth resolution is 4.5 mrad; scan interval is 30 s.”

Reviewer 2 Report
Dear authors,
I am satisfied with the changes you made to the manuscript and the answer you provided.
Anyway, please, have a look to these minor points below:
Line 40. The correct name of the author is “Del Ventisette”.
Line 86. Replace “though” with “through”.
Lines 105 and 116. Remove indentations and write “Write” with lower case as in lines after equations (5) and (6).
Lines 124 and 125. Use symbols for units: meters -> m
Line 129. Replace “stander uncertainly” with “standard uncertainty”.
Line 191. Replace “UnWrappe” with “UnWrapped”.
Page 7, first paragraph. Replace “-0.5 mm to 0.5 mm” with “-0.5 to 0.5 mm”. Do the same in the last paragraph of this page 7, in page 9, in page 11 line 26, lines 24 and 25.
Page 7. Table 2 caption. Replace “ps” with “PS”.
Page 9, last paragraph. Replace “An adjustable CR” with “an adjustable CR”.
Page 11, line 33. Replace “…in which of the…” with “…in which the…”
Page 11, line 36. Replace “,” with “.” Before “Details”
Page 11, line 46. Do not use contractions. Can’t -> cannot
Page 12, line 54. There is not congruency with subject in singular/plural and the verb.
Author Response
Dear,
Many thanks for the constructive and encouraging comments on our manuscript from yourself.We enclose a carefully revised manuscript and an item-by-item response according to the comments and suggestions made. The comments are included in “bold italics”, and our responses are in “regular” text.
We hope that these clarifications and revisions will now enable the paper to be accepted for publication in Journal of Remote Sensing, and look forward to hearing from you soon.
Yours sincerely,
Professor Sichun Long (on behalf of all authors)
Hunan University of Science and Technology, Xiangtan Hunan, China, 411201.
Response to Reviewer #2
Main Comments
please, have a look to these minor points below:
Line 40. The correct name of the author is “Del Ventisette”.
Line 86. Replace “though” with “through”.
Lines 105 and 116. Remove indentations and write “Write” with lower case as in lines after equations (5) and (6).
Lines 124 and 125. Use symbols for units: meters -> m
Line 129. Replace “stander uncertainly” with “standard uncertainty”.
Line 191. Replace “UnWrappe” with “UnWrapped”.
Page 7, first paragraph. Replace “-0.5 mm to 0.5 mm” with “-0.5 to 0.5 mm”. Do the same in the last paragraph of this page 7, in page 9, in page 11 line 26, lines 24 and 25.
Page 7. Table 2 caption. Replace “ps” with “PS”.
Page 9, last paragraph. Replace “An adjustable CR” with “an adjustable CR”.
Page 11, line 33. Replace “…in which of the…” with “…in which the…”
Page 11, line 36. Replace “,” with “.” Before “Details”
Page 11, line 46. Do not use contractions. Can’t -> cannot
Page 12, line 54. There is not congruency with subject in singular/plural and the verb.
Re: Thank you for your valuable suggestion. They have been modified in the revised version.
